# Cloning and Functional Characterization of Dihydroflavonol 4-Reductase Gene Involved in Anthocyanidin Biosynthesis of Grape Hyacinth

**DOI:** 10.3390/ijms20194743

**Published:** 2019-09-24

**Authors:** Hongli Liu, Qian Lou, Junren Ma, Beibei Su, Zhuangzhuang Gao, Yali Liu

**Affiliations:** 1College of Landscape Architecture and Arts, Northwest A & F University, Yangling 712100, China; liuhongli1221@sina.com (H.L.); subeibei1106@163.com (B.S.); gaoz4207@163.com (Z.G.); 2State Key Laboratory of Crop Stress Biology in Arid Areas, Northwest A & F University, Yangling 712100, China; louqian@nwsuaf.edu.cn; 3College of Horticulture, Northwest A & F University, Yangling 712100, China

**Keywords:** anthocyanin biosynthesis, dihydroflavonol 4-reductase, enzymatic reaction, flower color, grape hyacinth

## Abstract

Grape hyacinth (*Muscari* spp.) is a popular ornamental plant with bulbous flowers noted for their rich blue color. *Muscari* species have been thought to accumulate delphinidin and cyanidin rather than pelargonidin-type anthocyanins because their dihydroflavonol 4-reductase (DFR) does not efficiently reduce dihydrokaempferol. In our study, we clone a novel *DFR* gene from blue flowers of *Muscari. aucheri*. Quantitative real-time PCR (qRT-PCR) and anthocyanin analysis showed that the expression pattern of *MaDFR* had strong correlations with the accumulation of delphinidin, relatively weak correlations with cyanidin, and no correations with pelargonidin. However, in vitro enzymatic analysis revealed that the MaDFR enzyme can reduce all the three types of dihydroflavonols (dihydrokaempferol, dihydroquercetin, and dihydromyricetin), although it most preferred dihydromyricetin as a substrate to produce leucodelphinidin, the precursor of blue-hued delphinidin. This indicated that there may be other functional genes responsible for the loss of red pelargonidin-based pigments in *Muscari*. To further verify the substrate-specific selection domains of MaDFR, an assay of amino acid substitutions was conducted. The activity of MaDFR was not affected whenever the N135 or E146 site was mutated. However, when both of them were mutated, the catalytic activity of MaDFR was lost completely. The results suggest that both the N135 and E146 sites are essential for the activity of MaDFR. Additionally, the heterologous expression of *MaDFR* in tobacco (*Nicotiana tabacum*) resulted in increasing anthocyanin accumulation, leading to a darker flower color, which suggested that *MaDFR* was involved in color development in flowers. In summary, MaDFR has a high preference for dihydromyricetin, and it could be a powerful candidate gene for genetic engineering for blue flower colour modification. Our results also make a valuable contribution to understanding the basis of color variation in the genus *Muscari*.

## 1. Introduction

Anthocyanins are the most common pigment compounds responsible for orange, red, purple, and blue colors in many plant species [1,2]. Grape hyacinth (*Muscari* spp.) is a popular ornamental plant with bulbous flowers noted for their unusual blue color. Of the 52 grape hyacinth species currently known, over 90% produce flowers ranging in color from light blue to very dark blue (some varieties are almost black), although albino forms are also known. Mori (2002) analyzed the anthocyanin profiles of 13 grape hyacinth species and showed that flowers of these species accumulate delphinidin (Dp) and cyanidin (Cy) instead of pelargonidin (Pg) [3]. One possible reason for the absence of Pg-type anthocyanins and red flowers in the genus is that the Muscari dihydroflavonol 4-reductase (DFR) does not efficiently reduce dihydrokaempferol (DHK). In support of this, when the accumulation of Dp and Cy pigments is blocked due to a low expression level of DFR, resulting in blue flowering *Muscari. armeniacum* produces white flowers [4]. A similar case was found in the genera *Cymbidium* and *Petunia*, orange to brick red flowers are nonexistent because of their inability to accumulate Pg, which stems from the substrate specificity of their endogenous DFRs [5,6].

Dihydroflavonol 4-reductase (DFR) is a rate-limited enzyme in the biosynthesis of anthocyanins. It selectively catalyzes the reduction of three colorless dihydroflavonols (DHK, dihydroquercetin (DHQ], and dihydromyricetin (DHM]) to leucoanthocyanidins [7]. Subsequent steps are catalyzed by a series of downstream genes, yielding cultured anthocyanidins, orange Pg, reddish-purple Cy, and violet-blue Dp [8]. Depending on substrate specificity, DFR enzymes are divided into three types, which correlates with the amino acid 134. DFRs with asparagine (Asn), aspartic acid (Asp), or neither Asn nor Asp amino acid at position 134 (known as Asn-type, Asp-type, and non-Asn/Asp-type DFRs, respectively) utilize DHK, DHQ, and DHM as substrates, respectively [9,10,11]. This implies that different isoforms of DFR determine the content and ratios of Pg, Cy, and Dp, thus the final plants show different colors [12]. For example, *Petunia* and *Cymbidium* species lack brick-red to orange colored varieties because of the absence of Pg-type anthocyanins, as their DFRs do not use DHK as a substrate [5,6]. Similarly, DFRs in *Iris* and *Gentian* species prefer DHM to DHK as a substrate, and DHM is converted to blue Dp derivatives more readily than to orange Pg derivatives. The substrate specificity of DFR also plays a role in molecular breeding. Upon introducing maize (*Zea mays*) *DFR* into white-flowered varieties of petunia lacking enzyme, the flowers of transgenic plants accumulate non-native Pg, resulting in a novel brick red-flowered variety [13]. To produce blue flowers in rose (*Rosa* spp.), the gene encoding DHQ-specific *DFR* was deleted, and the *Iris DFR* encoding gene, which has a high preference for DHM, was introduced together with the *Viola* flavonoid 3′, 5′-hydroxylase (*F3′5′H*) encoding gene [14]. Thus, it is reasonable to speculate that such substrate preferences of *DFR*s are relatively unconstrained among plant species, leading to the accumulation of different anthocyanins among species, thereby exhibiting a limited color range.

Here, a novel *DFR* gene was isolated from *Muscari. aucheri*. Quantitative real-time PCR (qRT-PCR) analysis shows that changes in *MaDFR* expression are highly correlated with the accumulation of Dp derivatives. Enzyme activity analyses show that DHM is the preferred substrate for MaDFR. The heterologous expression of *MaDFR* in tobacco increased the accumulation of anthocyanins, leading to a darker variation in flower color. These results suggest that *MaDFR* is associated with color development and is helpful for understanding anthocyanin biosynthesis in grape hyacinth.

## 2. Results

### 2.1. Cloning and Sequence Analysis of MaDFR

The full-length cDNA of *MaDFR* (GenBank accession no. MK937098) from grape hyacinth (*M. aucheri,* ‘Dark eyes’) is 1101 bp in length (Appendix A) and encodes a protein of 366 amino acids (Appendix A) and 40.93 kDa molecular weight. The deduced MaDFR amino acid sequence showed high similarity with the DFR proteins from *Hyacinthus orientalis* (87.19%), *Iris × hollandica* (67.65%), and *Antirrhinum majus* (48.65%) (Figure 1A). The multiple amino acid sequence alignment also revealed the highly conserved NADPH-binding motif (VTGAAGFIGSWLIMRLLERGY) [15] and the substrate-binding domain (T133–K158) [11] in the MaDFR sequence (Figure 1A). These results indicate that *MaDFR* can be classified into the short chain dehydrogenase/reductase (SDR) superfamily of proteins, which catalyze the reduction of three colorless dihydroflavonols to leucoanthocyanidins in the anthocyanin synthetic pathway.

To further investigate the relationship between the MaDFR protein and other known DFRs, a phylogenetic tree was generated with the maximum likelihood method using MEGA 7.0. Phylogenetic analysis divided the DFR family into two groups: monocot and eduicot (Figure 1B). The MaDFR protein clustered in the monocot group and was most similar to the DFRs from *H. orientalis*, *Allium cepa*, *Freesia hybrid*, and *Iris × hollandica*. Interestingly, most of these species produce blue flowers. These results show that *MaDFR* is probably involved in the synthesis of blue pigments in grape hyacinth flowers.

### 2.2. Anthocyanin Accumulation and MaDFR Expression in Grape Hyacinth

To gain insight into the regulation of *MaDFR*, anthocyanin accumulation and *MaDFR* expression were analyzed in the inflorescence of grape hyacinth at five different developmental stages (S1–S5) (Figure 2A). The accumulation of anthocyanin showed significant spatial and temporal specificity. No anthocyanin was detected in root, bulb, or leaf tissues of grape hyacinth; anthocyanins started to accumulate in the flowers at stageS1, peaked at stage S4, and decreased at stage S5 (Figure 2C). In flowers, Dp was the major anthocyanin, and its accumulation pattern was highly correlated with the total anthocyanin content (Figure 2D). Cyanidin also showed relatively high accumulation in grape hyacinth flowers (Figure 2E); however, the amount of Pg was negligible (Figure 2F). Gene expression analysis revealed very low levels of *MaDFR* transcripts in roots, bulbs, and leaves. In flowers, the expression level of *MaDFR* increased dramatically from stage S1 to stage S3, reaching a peak at stage S3, followed by a decline until stage S5 (Figure 2B). The expression of *MaDFR* showed the strongest correlation with the accumulation pattern of Dp (*r* = 0.850), followed by that of Cy (*r* = 0.632), and no correlation with the accumulation of Pg (*r* = 0.234). The correlation of Pg, Cy and Dp with anthocyanin biosynthetic gene expression levels is shown in the Appendix A. Overall, these results show a tissue-specific expression pattern for *MaDFR*, which is highly correlated with anthocyanin accumulation in flowers.

### 2.3. Functional Expression in E. coli and Enzyme Assay

To characterize the substrate-specific function of *MaDFR*, we cloned the coding sequence of *MaDFR* and three mutants (*MaDFRa*, *MaDFRb*, and *MaDFRc*) into pET-28a expression vector (Figure 3) and expressed the recombinant proteins in *E. coli* strain BL21. The recombinant proteins were separated followed by sodium dodecyl sulfate polyacrylamide gel electrophoresis (SDS-PAGE) (Figure 3A and Appendix A).The MaDFR protein was identified at the predicted molecular mass of 41 kDa by Western blotting (Figure 3B and Appendix A). For the enzyme activity assay, crude bacterial protein extracts were used, as DFR proteins are reported to be unstable during purification [16]. The activity of MaDFR on the three dihydroflavonols (DHK, DHQ, and DHM) was characterized by HPLC-MS analysis (Figure 3D). Leucoanthocyanidins are unstable; therefore, the reaction was conducted chemically using acidic alcohols to generate the stable corresponding anthocyanidins (Figure 3D) [17]. Peak1 appeared when DHK was contained in the reaction solution, and the corresponding products showed a brick red color (Figure 3D). The compounds in peak1 had the same retention time as the standard of Pg, with major fragments at *m*/*z* ratios of 271.2, 340.1, and 183.2, as in the Pg standard (Figure 3C,D). This suggests that peak1 contained Pg. Peak2 was produced when DHQ was used as the substrate, and the corresponding products showed a red color (Figure 3D). Compounds in peak2 eluted at the same retention time and mass spectrum, with major fragments at m/z ratios of 287.2, 241.0, and 213.1, as in the Cy standard (Figure 3C,D), Therefore, peak2 was annotated as Cy. Peak3 was produced when DHM was used as substrate, and the corresponding products showed a violet color (Figure 3C). Peak3 matched the retention time and mass spectrum (*m*/*z* ratios of 303.1, 285.3, and 259.2) of the Dp standard (Figure 3C,D), indicating that peak3 represented Dp. In summary, these results indicate that MaDFR efficiently catalyzes the reduction of all three dihydroflavonols (DHK, DHQ, and DHM) to the corresponding leucoanthocyanins (leucopelargonidin, leucocyanidin, and leucodelphinidin, respectively). All enzymatic assays were carried out under the same conditions. Quantification analysis of the reaction products indicated that MaDFR showed the strongest preference for DHM as a substrate, followed by DHQ, and lastly DHK (Table 1). Among the mutant MaDFR proteins, MaDFRa and MaDFRb catalyzed all three substrates, while MaDFRc had no catalytic activity (Figure 4). These results suggest that at least one of the residues (asparagine (N135) and glutamic acid residues (E146)) is necessary for the catalytic activity of the MaDFR enzyme.

### 2.4. Enhanced Production of Anthocyanins by Introducing MaDFR into Transgenic Tobacco

To investigate the function of *MaDFR*, a binary vector harboring the *MaDFR* gene under the control of the cauliflower mosaic virus *35S* promoter was transformed in tabacum. We obtained 11 independent transgenic lines with different colored flowers by gene-specific PCR analysis and did not determine the copy number of T-DNA insertions in the chosen lines (data not shown). Three transgenic lines with dark red flowers were selected for further analysis. Compared with the control (CK) transgenic line (pale pink flowers) carrying the empty vector, the *p35SMaDFR* transgenic tobacco lines in the T3 generation showed more pigmentation (Figure 5A). The expression level of *MaDFR* in transgenic tobacco lines was confirmed by qPCR. The results show that the transcript levels in transgenic plants of *MaDFR* were higher than in the CK line (Figure 5B). Anthocyanins were extracted from the corollas of the three transgenic lines and quantified using as spectrophotometer. Results show that anthocyanin levels in the flowers of *MaDFR* overexpression line 3 (OE3) were three-fold higher than in CK (Figure 5C), indicating that the enzyme encoded by *MaDFR* effectively increased anthocyanin accumulation in vivo by interacting with the endogenous tobacco anthocyanin biosynthesis pathway enzymes.

In order to investigate the effect of *MaDFR* on endogenous tobacco genes involved in flavonoid synthesis, we carried out qPCR analysis of endogenous genes including *NtCHS*, *NtCHI*, *NtF3H*, *NtF3’H*, *NtF3’5’H*, *NtDFR*, *NtANS*, and *NtUFGT,* and *three regulatory genes*, *NtAn2*, *NtAN1a*, and *NtAN1b* (Figure 6A). The expressions of *NtANS* and *NtUFGT* were significantly higher in transgenic tabacum lines overexpressing *MaDFR* than in CK, whereas the expressions of the other genes were either slightly higher (*NtAn2*, *NtAN1a*, and *NtAN1b*) or slightly lower (*NtCHI*) in the transgenic lines than in CK (Figure 6B,C).

## 3. Discussion

In the flavonoid biosynthesis pathway, dihydroflavonol-4-reductase (*DFR*) genes play an important role in the formation of anthocyanin pigments. *DFR* genes belong to the short chain dehydrogenase reductase (SDR) superfamily, which have a highly conserved NADPH-binding domain, and a substrate-binding domain in plants [18,19]. Here a novel *DFR* gene was cloned from grape hyacinth, The NADPH-binding domain and the substrate-binding domain were found in MaDFR through amino acid alignment. A phylogenic tree showed high similarity between *MaDFR* and other characterized DFRs, indicating that it belongs to the monocot DFR family and has catalytic properties. In order to study the functions of *MaDFR*, the expression patterns were tested temporally and spatially. In different tissues and flower development stages of grape hyacinth, its expression level increased with the color development. The expression pattern of *MaDFR* was correlated with the accumulation of total anthocyanins, and the expression of *MaDFR* showed the strongest correlations with the accumulation pattern of Dp, relatively weak correlations with Cy, and no correlations with Pg (Figure 7). These results indicated that *MaDFR* was related to the flower color development of grape hyacinth.

Previous studies demonstrated that *DFR*s can be divided into three types according to the 134th amino acid, i.e., Asn-type DFRs (the 134th amino acid is asparagine residue(n)), Asp-type DFRs (the 134th amino acid is aspartic acid (d)), and non-Asn/Asp-type DFRs (the 134th amino acid is neither n nor d) [7,11]. In most plant species, the 134th amino acid is n or d, and the 145th is glutamic acid (E). MaDFR belonged to Asn-type DFRs (Figure 2A), and it could catalyze DHK, DHQ and DHM, which was consistent with previous reports [6,11]. However, not all Asn-type DFRs can utilize all the three dihydroflavonols as substrates. For example, the *DFR* from purple-fleshed potato (IbDFR) belongs to Asn-type DFRs, but only DHK could be catalyzed [20]. FhDFR2 could utilize DHM and DHQ as substrates but could not reduce DHK to leucopelargonidin [21]. In this study, MaDFR preferentially catalyzeds DHM and could use all the three dihydroflavonols as substrates, which provides a new insight to understand anthocyanin biosynthesis in grape hyacinth, and it may be an ideal candidate gene to specifically engineer the biosynthesis of delphindin-type anthocyanins.

Johnson et al. found that DFR substrate specificity could be changed by altering the 134th and 145th amino acid [11]. Furthermore, the analysis of point mutations of the 135th and 146th amino acids of MaDFR (corresponding to the 134th and 145thamino acids of GhDFR, respectively) (Figure 1) were analyzed. The results showed that whether the 134th or 145th amino acid was mutated to polar leucine, MaDFR could still catalyze three dihydroflavonols. Only when the 134th and 145th amino acids were simultaneously mutated to polar leucine, DFR lost its catalytic activity (Figure 4). These results indicated that there might be other amino acid binding sites influencing the substrate specificity of MaDFR. The 133th amino acid of Strawberry DFR2 (corresponding to the gerbera DFR of the 134th amino acid) is asparagine (N), and DFR1 in this corresponding position is alanine (A) [22]. In vitro enzymatic analysis indicated that strawberry DFR1 tended to use DHK as a substrate, while DFR2 could utilize DHQ and DHM as substrates, but not DHK [22]. In summary, the substrate specificity of DFR is not completely determined by the correspondence to the 134th amino acid of gerbera DFR.

In order to further investigate the potential functions of the *MaDFR* gene, *MaDFR* was introduced into tobacco, and an increase was observed in Cy-type anthocyanins and dark-red flowers in the transgenic lines (Figure 5A). Interestingly, overexpression of *MaDFR* in transgenic tobacco enhanced the biosynthesis of anthocyanins by increasing the expression of downstream genes (*NtANS* and *NtUFGT*) and regulatory genes (*NtAN2*, *NtAN1a*, *NtAN1b*) (Figure 6B,C). NtAN2 and NtAn1a/b are a R2R3-Myb transcription factor and two bHLH transcription factor genes, respectively, which regulate tobacco anthocyanin synthesis [23,24], thus increasing the total metabolic flux. Unexpectedly, no Dp-type pigments or bluer flowers were produced. This may be because the host plant is incapable of producing Dp because of DHM deficiency, and therefore can produce only Cy-type anthocyanins. A similar phenomenon leading to an unexpected product has been observed in *Nierembergia*. Suppression of the endogenous *F3′5′H* gene and overexpression of a rose *DFR* led to the production of white flowers instead of the desired red flower because of its weak *F3′H* activity and strong *FLS* activity [25]. Additionally, overexpression of maize transcription factors in tomato (*Solanum lycopersicum*) significantly upregulated several flavonoid genes and increased the accumulation of kaempferol in the fruit, but it was not enough to induce the activity of *F3′5′H*, which is necessary for anthocyanin synthesis [26]. In rose, because the endogenous *DFR* gene leads the metabolic flux toward Cy, blue-flowered rose varieties exclusively containing Dp could not be generated by introducing the viola *F3′5′H* gene only [27].

In nature, it is rare for a single plant genus to contain varieties with a full spectrum of flower colors because of the constraints imposed by the gene pool. For instance, the *DFR* genes of *Iris* and *Gentiana*, which mainly accumulate Dp rather than Pg, have been correlated with the absence of brick-red flowers [27]. It seems that a single *DFR* gene is an ideal target for determining flower color for plant breeders. Firstly, as a ‘late gene’, *DFR* does not disturb plant development too much. Secondly, *DFR* is more important for pigmentation than other late genes, such as those encoding anthocyanin glycosylation, methyltransferase or acyltransferase, which only modify the hue of the flower color. Nevertheless, it seems that such flower color preference is relatively unconstrained in the genus because it can be achieved in many ways. The inactive *F3′5′H* gene is associated with the absence of blue flowers in the three best-selling fresh cut flowers: roses, carnations, and chrysanthemum [27,28]. Another common reason for limited flower color is strong competition between rival genes or metabolic pathways. Strong activity of *F3′5′H* and *FLS* in *Nierembergia* drives the metabolic flux to Dp-based anthocyanins and colorless flavonols, thus generating violet-blue or white flowers. Downregulation of the *F3′5′H* and *FLS* genes together with the expression of a rose *DFR* redirects the Dp pathway to Cy, thus yielding transgenic lines with novel pink flowers [25]. In *Muscari*, although the hypothesis that a single DFR is responsible for the limited color of the flowers is attractive, it is not the only reason. The presence of Pg derivatives in blue petals and the catalytic activity of MaDFR for DHK indicated that there must be the anthocyanin biosynthetic pathway based on the red Pg in the genus, hinting that other functional genes are responsible for the loss of red Pg-based pigments in grape hyacinth. In order to find more blue color-determining genes, further explanation of downstream genes and regulatory genes are needed.

## 4. Materials and Methods

### 4.1. Plant Materials and Growth Conditions

Grape hyacinth (*M. aucheri* ‘Dark Eyes’) and tobacco (*Nicotiana. tabacum* cv. ‘NC89′) were used in this study. Grape hyacinth plants were grown outdoors in Northwest A&F University, which is located in Yangling, China. Tobacco plants for transformation were cultured in a growth chamber (25 °C) with a 8 h dark/16 h light photoperiod (13,000 lx). All samples were frozen without delay in liquid nitrogen and stored at −80 °C before use.

### 4.2. Cloning of the MaDFR Gene

An *MaDFR* gene full-length sequence was obtained by screening the transcriptome data of grape hyacinth [29]. Total RNA was extracted from the petals of grape hyacinth using the Total RNA Kit (Omega, Norcross, GA, USA) and used as the initial material for gene cloning. The full-length gene coding sequence was cloned by designing specific primers (Appendix A). PrimeSTAR^™^ HS DNA Polymerase (Takara Biotechnology, Dalian, China) was used to clone the cDNA following the instructions. The obtained MaDFR cDNA sequence was confirmed using the BLAST program (NCBI, USA).

### 4.3. Sequence Alignment and Phylogenetic Analysis

The amino acid sequence alignment of DFR and other DFR proteins from GenBank was analyzed using CLC Sequence Viewer 8.0 (CLC bio, Aarhus, Denmark). A phylogenetic tree was constructed using MEGA7.0 (Kumar, Stecher, Li, Knyaz, and Tamura 2018) with the maximum likelihood method (Figure 1).

### 4.4. Expression Analysis of MaDFR by qRT-PCR

Total RNA was extracted from different tissues of grape hyacinth, including roots, bulbs, and leaves, and from the flowers at five developmental stages using the Total RNA Kit (Omega, Norcross, GA, USA). The five stages (S1–S5) of petals of grape hyacinth were divided according to the petal pigmentation [30] (Figure 2A). Next, 1 μg of RNA was used to synthesize cDNA by using the PrimeScript™ RT reagent Kit with gDNA Eraser (Takara Biotechnology, Dalian, China), following the manufacturer’s instructions. Then, qRT-PCR was carried out on the iQ5 RT-PCR instrument (Bio-Rad, Hercules, CA, USA) using AceQqPCR SYBR^®^ Green Master Mix (Vazyme, Nanjing, China). The *MaActin* gene was chosen as the internal control gene for *Muscari.* Spp, and the *NtTubA1* gene was used as the internal control gene for tabacum samples. Primers used for qRT-PCR are listed in Appendix A. The relative expression of *MaDFR* was quantified using the 2^−^^△△*C*t^ method [31]. Using the Correlationplot Rectanglel module of Qi MetA software (Qiji Biotechnology Co, Ltd, Beijing, China), the *r* value of MaDFR expression and anthocyanin accumulation of the five stages of flower development were recalculated using the Pearson method, not including the roots, bulbs, and leaves. All analyses were conducted with three independent experiments.

### 4.5. Plasmid Construction

Mutant variants of the *MaDFR* genes (*MaDFRa*, *MaDFRb*, and *MaDFRc*), each containing a point mutation, were generated as described previously [11] (Figure 7). The wild-type *MaDFR* gene and all three mutant variants (*MaDFRa*, *MaDFRb*, and *MaDFRc*) were subcloned into the pET28a-His vector separately, the location of the His tag is at the C-terminal of the pET28a-His vector. Primers used for vector construction are listed in Appendix A.

### 4.6. Heterologous Expression of MaDFR and in Vitro Enzyme Assay

The *Escherichia coli* strain BL21 (DE3) was transformed with the empty pET28a-His vector, recombinant *MaDFR, MaDFRa, MaDFRb,* and *MaDFRc*. The transformants were incubated in Luria–Bertani (LB) medium overnight including 50 μg/mL kanamycin at 37 °C and 280 rpm agitation. On the next day, a 1:50 dilution of the culture was prepared in fresh LB medium containing 50 μg/mL kanamycin and incubated at 37 °C and 280 rpm agitation, until the optical density of the culture at 600 nm absorbance (OD_600_) reached a value of 0.8. To induce recombinant protein production, 0.2 mM of isopropyl-β-d-thiogalactopyranoside (IPTG) was added to the culture and incubated at 16 °C for 20 h. After induction, the cells were centrifugated at 12,000 rpm for 10 min to obtain the bacterial pellet, which was resuspended in phosphate-buffered saline (PBS; pH 7.4); next, it was disrupted using a Noise Isolating Chamber (Scientz, Ningbo, China) for 15 min, with an output power of 200 W. After that, the mixture was collected at 12,000 rpm for 20 min, and the supernatant containing crude proteins was collected. A quantity of 50 μL of crude proteins was used for SDS-PAGE and immunoblotting with a specific antibody against His (the antibody sequence for HHHHHH, TransGen Biotech, Beijing, China); the goat anti-mouse IgG (H+L) (TransGen Biotech, Beijing, China) was used to recognize the mouse monoclonal anti-His antibodies, and the antibodies were used following the instructions. The experimental operation was according to a method reported by predecessors [32].

Substrate specificities of MaDFR protein was analyzed as depicted formerly [16]. All three substrates (DHK, DHQ, and DHM) were dissolved in methanol (10 μg/mL; Sigma). A 500 μL reaction mixture containing 70 μL 10 mM Tris-HCl (pH 7.0), 50 μL 20 mM nicotinamide adenine dinucleotide phosphate (NADPH), 10 μL substrate, and 370 μL crude protein extract containing MaDFR, MaDFRa, MaDFRb and MaDFRc, respectively, was cultivated at 37 °C for 1h. An equal volume of n-butanol:HCl (95:5, *v*/*v*) was added because leucoanthocyanidin is unstable, and the reactions were cultivated at 95 °C for 1 h to form anthocyanidins. The n-butanol layer was transferred to a new 2.0 mL tube and evaporated using a stream of nitrogen gas. The residue was dissolved in 50 μL methanol and used directly for liquid chromatography-mass spectrometry (LC-MS) analysis.

### 4.7. Analysis of MaDFR Reaction Products

The reaction products of MaDFR MaDFRa, MaDFRb, and MaDFRc were identified by high-performance liquid chromatography (HPLC) analysis using the Agilent1100 HPLC system (Agilent, USA) at a detection wavelength of 530 nm. The TC-C18 column (4.6 mm × 250 mm, 5 μm) was kept at 30 °C. The samples were eluted using 0.1% formic acid (solvent A) as well as acetonitrile (solvent B), according to a program of gradient elution described previously [33].

The MaDFR reaction products were quantified by HPLC electrospray ionization tandem mass spectrometry (HPLC-ESI-MS/MS) analysis using the Q-TRAP5500 tandem mass spectrometer (AB SCIEX, Washington, DC, USA), equipped with ESI and controlled by Analyst software (version 1.6.1, AB SCIEX, Washington, DC, USA). First, the mass-to-charge (*m*/*z*) ratios of parent ions were confirmed with a scanned area of approximately 50–200 Da/s and scanned speed of 200 Da/s. Second, the m/z ratios of daughter ions were measured using the following steps: (1) setting the initial value (the initial value of collision energy(CE) was 5 eV); (2) taking 5 eV as a step length to adjust the value of CE manually, which is appropriate if the strength of parent ions is one-third of that of the base peak; (3) establishing the multiple reaction detection ion pair according to previously selected daughter ions and parent ions, (optimized CE and declustering potential (DP) with Ramp).

### 4.8. Stable Transformation of Tobacco

The coding sequence of *MaDFR* was cloned into the pCAMBIA2300 expression vector to produce the recombinant plasmid *p2300-35SMaDFR*. Next, the recombinant vector *p2300-35sMaDFR* was introduced into *Agrobacterium tumefaciens* strain GV3101 by electroporation, the empty vector as a control. The transformation of tobacco plants by means of leaf disk transformation-regeneration was conducted as described previously [33]. Transgenic plants were identified by PCR amplification using genomic DNA isolated from young leaves [34] along with sequence-specific primers (2300-F and 2300-R; Appendix A).

### 4.9. Quantification of Anthocyanin Content

Total anthocyanins were extracted from different tissues of grape hyacinth and flowers of transgenic tobacco lines and quantified as described previously [35,36]. Samples of grape hyacinth were mashed into a fine powder in liquid nitrogen. To extract anthocyanins, methanol:water:formic acid:TFA solution (70:27:2:1, *v*/*v*) was added to the sample, which was then cultivated in the dark at 4 °C for 24 h [37]. Hydrolysis and HPLC of anthocyanins were performed, following the previous description [4]. The anthocyanin content of grape hyacinth was determined using delphinidin 3-*O*-glucoside (Sigma, St. Louis, MO, USA) equivalents. Total anthocyanin content of tobacco was measured with a spectrophotometer (UV2600, Shimadzu, Kyoto, Japan) and calculated using the following equation [38]:(1)$$Total\;anthocyanin\;content_{Tobacco}=\frac{A_{530}-\left(0.25\;\times \;A_{657}\right)}{Fresh\;weight}$$

All samples were analyzed in three biological replicates.

### 4.10. Statistical Analysis

Statistical analysis of data was carried out using analysis of variance (ANOVA) with SPSS Statistics. Significant differences were determined using the Tukey–Kramer test at *p* < 0.05.

## 5. Conclusions

The newly identified *MaDFR* gene was associated with color development, which was highly correlated with the accumulation of Dp derivatives. Moreover, the MaDFR enzyme could utilize all the three types of dihydroflavonols as substrates, and it most preferred dihydromyricetin as a substrate to produce leucodelphinidin. This finding might provide a new insight to understand the anthocyanin biosynthesis pathway in grape hyacinth.

## Figures and Tables

**Figure 1 ijms-20-04743-f001:**
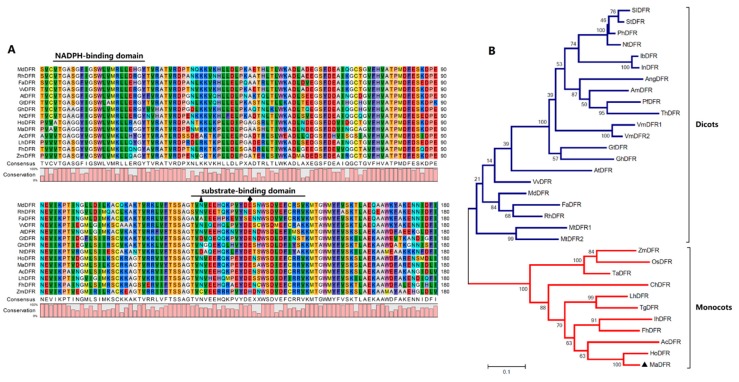
Alignment of the protein sequence and phylogenetic tree analysis of MaDFR. (**A**). Multiple sequence alignment of the predicted amino acid sequences of MaDFR and dihydroflavonol 4-reductase (DFR) proteins from other plant species. The alignment was generated using CLC Sequence Viewer 8.0. The NADPH-binding and substrate-binding domains of DFR are indicated. Substrate specificity of DFRs is particularly associated with amino acids at positions 134 and 145, indicated by the black triangle and black rhombus, respectively. DFR conserved amino acid residues are highlighted in different colors. (**B**). Phylogenetic analysis of MaDFR with other DFR proteins. The phylogenetic tree was generated using the maximum likelihood method in MEGA 7.0 software. Numbers at each interior branch indicate the bootstrap values of 1000 replicates. The bar indicates a genetic distance of 0.1. Plant species and GenBank accession numbers of their DFR proteins used for phylogenetic analysis are: *Muscari aucheri* MaDFR (MH636605, marked with a black triangle), *Solanum lycopersicum* SlDFR (CAA79154.1), *Solanum tuberosum* StDFR (AF449422), *Petunia* × *hybrida* PhDFR (AF233639), *Nicotiana tabacum* NtDFR (NP_001312559.1), *Ipomoea batatas* IbDFR (HQ441167), *Ipomoea nil* InDFR (AB006793), *Angelonia angustifolia* AngDFR (KJ817183), *Antirrhinum majus* AmDFR (X15536), *Perilla frutescens* PfDFR (AB002817), *Torenia hybrid* ThDFR (AB012924), *Vaccinium macrocarpon* VmDFR1 (AF483835), *Vaccinium macrocarpon* VmDFR2 (AF483836), *Gentiana triflora* GtDFR (D85185), *Gerbera hybrid* GhDFR (Z17221), *Arabidopsis thaliana* AtDFR (AB033294), *Vitis vinifera* VvDFR (Y11749), *Malus domestica* MdDFR (AAO39816), *Fragaria × ananassa* FaDFR (AF029685), *Rosa hybrid* RhDFR (D85102), *Medicago truncatula* MtDFR1 (AY389346), *Medicago truncatula* MtDFR2 (AY389347), *Zea mays* ZmDFR (Y16040), *Oryza sativa* OsDFR (AB003495), *Triticum aestivum* TaDFR (AB162139.1), *Cymbidium hybrid* ChDFR (AF017451), *Lilium hybrid* LhDFR (AB058641), *Tulipa gesneriana* TgDFR (BAH98155.1), *Iris × hollandica* IhDFR (BAF93856.1), *Freesia hybrid* FhDFR (KU132389), *Allium cepa* AcDFR (AY221250.2), and *Hyacinthus orientalis* HoDFR (AFP58815.1).

**Figure 2 ijms-20-04743-f002:**
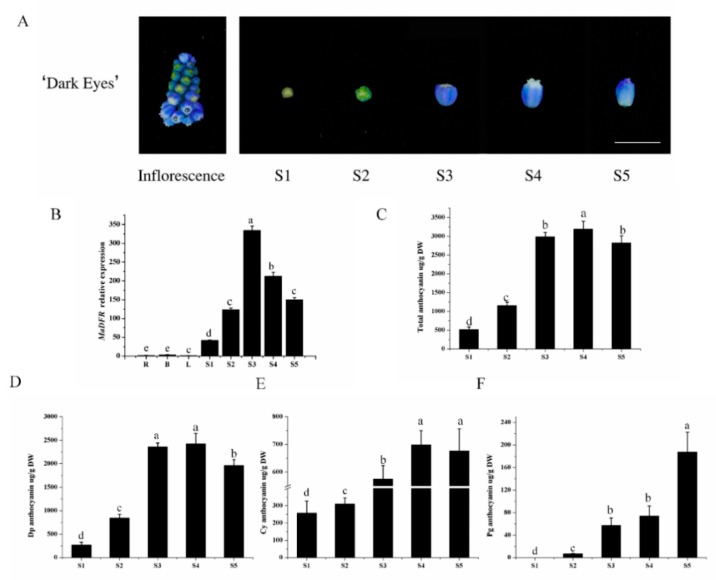
Anthocyanin content and *MaDFR* expression profiles in grape hyacinth. (**A**) The inflorescence and petals at five developmental stages. Flower development is divided into five stages: S1, newly formed flower buds with a little color; S2, development of petals with some color; S3, completely colored but closed petals; S4, completely colored and open petals; S5, flower senescence. Scale bars = 5 mm. (**B**) The expression profile of *MaDFR*, *MaDFR* expression in roots (R), bulbs (B), leaves (L) and flowers at five developmental stages. (**C**). Accumulation of total anthocyanins at five developmental stages in grape hyacinth. (**D**). Accumulation of delphinidin (Dp) at five developmental stages in grape hyacinth. (**E**). Accumulation of cyaniding (Cy) at five developmental stages in grape hyacinth. (**F**). Accumulation of pelargonidin Pg at five developmental stages in grape hyacinth. DW: Dry weight. Data represent the means ± SEs of three independent experiments.

**Figure 3 ijms-20-04743-f003:**
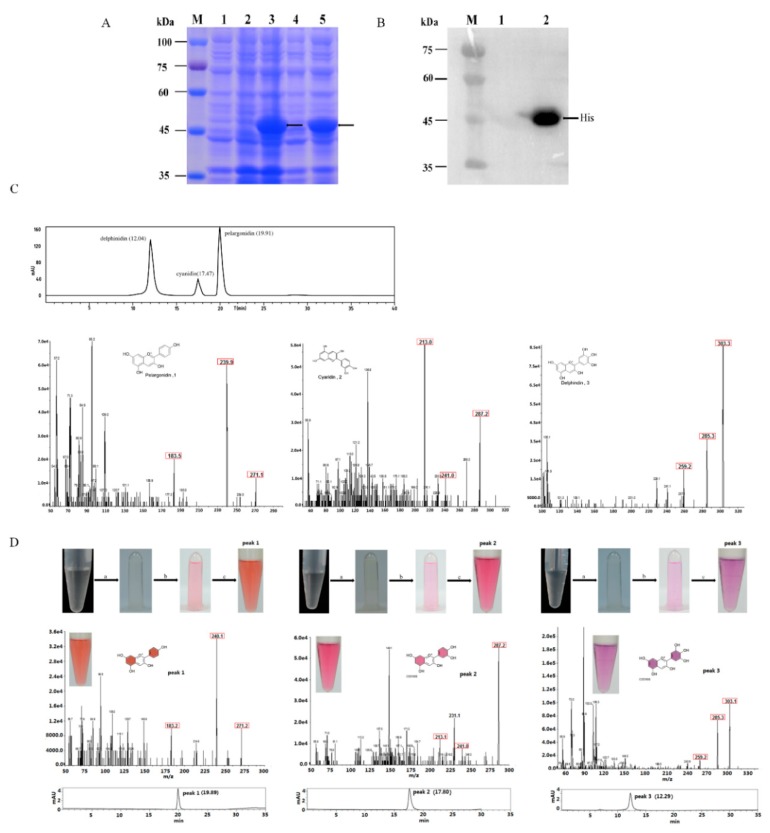
SDS-PAGE analysis and enzyme activity assay of the His-tagged MaDFR protein. (**A**) Coomassie brilliant blue stained polyacrylamide gel. M, protein marker. Lane 1, total protein extract of *E. coli* BL21 (DE3) harboring the empty pET-28a expression vector. Lane 2, total protein extract of *E. coli* BL21 (DE3) containing the expression plasmid pET-28a-MaDFRbefore induction. Lane 3, total protein extract of *E. coli* BL21 (DE3) containing the expression plasmid pET-28a-MaDFR after induction. Lane 4, the bacterium containing crude proteins extract of *E. coli* BL21 (DE3), containing the expression plasmid pET-28a-MaDFR after induction. Lane 5, the supernatant containing crude protein extract of *E. coli* BL21 (DE3), containing the expression plasmid pET-28a-MaDFR after induction. (**B**) Western blot analysis of the His-tagged DFR protein. M, Protein marker. Lane 1, negative control. Lane 2, MaDFR protein. (**C**) HPLC−MS/MS analysis of the standards of pelargonidin, 1; cyanidin, 2; and delphinidin, 3. The concentrations of these standards was 1000 ng/ mL. (**D**) HPLC−MS/MS analysis of MaDFR reaction products. Enzymatic reaction process of three substrates, the substrate of peak1 is dihydrokaempferol (DHK), the substrate of peak2 is dihydroquercetin (DHQ), and the substrate of peak3 is dihydromyricetin (DHM). a. A 500 µL reaction mixture containing 70 µL (10 mM) Tris-HCl (pH 7.0), 50 µL (20 mM) NADPH, 370 µL crude protein extract containing MaDFR and 10 µL of DHK, DHQ and DHM, respectively, was incubated at 37 °C for 1 h; next, an equal volume of n-butanol:HCl (95:5, *v*/*v*) was added. b. The reactions were incubated at 95 °C for 1 h to form anthocyanidins. c. the n-butanol layer was evaporated using a stream of nitrogen gas and the residue was dissolved in 100 µL of methanol. Numbers in brackets indicate retention times. Numbers in red boxes indicate the major fragment masses.

**Figure 4 ijms-20-04743-f004:**
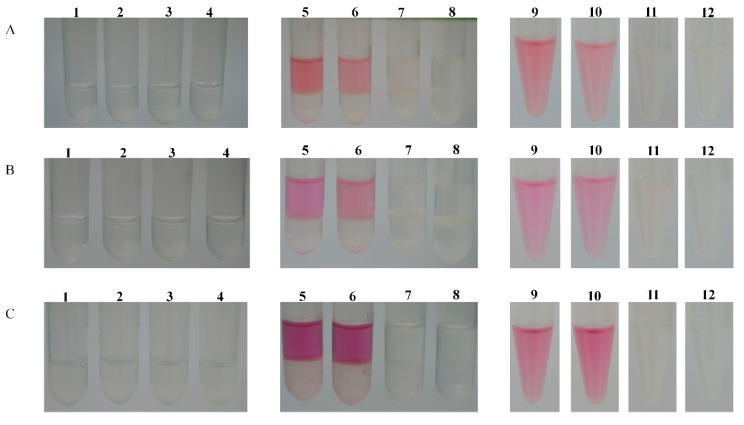
Enzyme assay of the MaDFRa, MaDFRb and MaDFRc proteins. (**A**). Enzymatic reaction process of the substrate DHK. 1, 2, 3 and 4; A 500 µL reaction mixture containing 70 µL (10 mM) Tris-HCl (pH 7.0), 50 µL (20 mM) NADPH, 10 µL DHK and 370 µL crude protein extract containing MaDFRa, MaDFRb, MaDFRc and the empty pET-28a, respectively, was incubated at 37 °C for 1 h. 5, 6, 7 and 8: An equal volume of n-butanol:HCl (95:5, *v*/*v*) was added, respectively, and then the reactions were incubated at 95 °C for 1 h to form anthocyanidins. 9, 10, 11 and 12: The n-butanol layer was evaporated using a stream of nitrogen gas and the residue was dissolved in 100 µL of methanol, respectively. (**B**). Enzymatic reaction process of the substrate DHQ. 1, 2, 3 and 4: A 500µL reaction mixture containing 70 µL (10 mM) Tris-HCl (pH 7.0), 50 µL (20 mM) NADPH, 10 µL DHQ and 370 µL crude protein extract containing MaDFRa, MaDFRb, MaDFRc and the empty pET-28a, respectively, was incubated at 37 °C for 1 h. 5, 6, 7 and 8: An equal volume of n-butanol:HCl (95:5, *v*/*v*) was added, respectively, the reactions was incubated at 95 °C for 1h to form anthocyanidins. 9, 10, 11 and 12: The n-butanol layer was evaporated using a stream of nitrogen gas, and the residue was dissolved in 100 µL of methanol, respectively. (**C**). Enzymatic reaction process of the substrate DHM. 1, 2, 3 and 4: A 500 µL reaction mixture containing 70 µL (10 mM) Tris-HCl (pH 7.0), 50 µL (20 mM) NADPH, 10 µL DHM and 370 µL crude protein extract containing MaDFRa, MaDFRb, MaDFRc and the empty pET-28a, respectively, was incubated at 37 °C for 1 h. 5, 6, 7 and 8: An equal volume of n-butanol:HCl (95:5, *v*/*v*) was added, respectively, and then the reactions were incubated at 95 °C for 1 h to form anthocyanidins. 9, 10, 11 and 12: The n-butanol layer was evaporated using a stream of nitrogen gas and the residue was dissolved in 100 µL of methanol, respectively.

**Figure 5 ijms-20-04743-f005:**
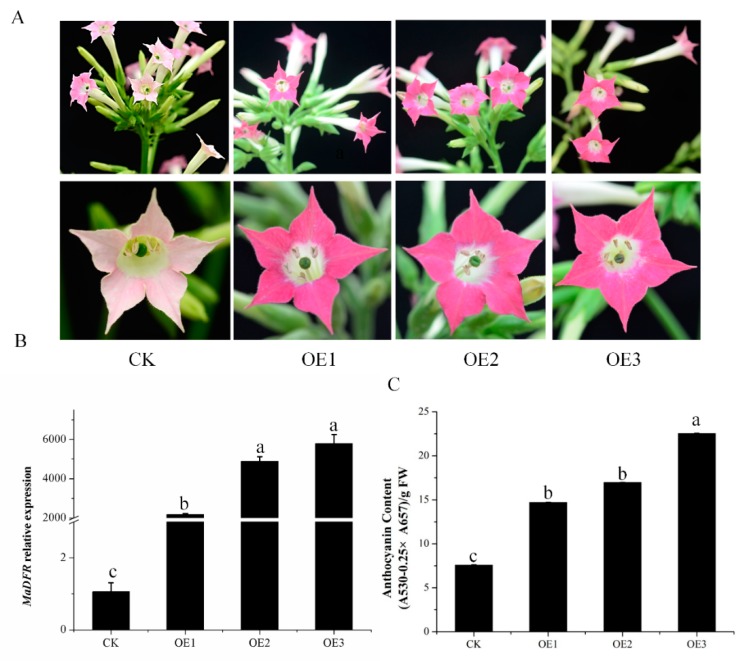
Characterization of transgenic tobacco lines overexpressing *MaDFR*. (**A**) Phenotypic analysis of *MaDFR* overexpression (OE1, OE2 and OE3) lines and control (CK) line transformed with pCAMBIA2300 empty vector. Heterologous expression of *MaDFR* in tobacco resulted in a distinct phenotypic change in petal color. (**B**) Relative expression of *MaDFR* in tobacco petals of three independent OE lines determined using qRT-PCR. *NtTubA1* was used as the internal control. (**C**) HPLC analysis of anthocyanin levels (mg/g fresh weight (FW)) in the petals of the CK line and OE lines. Data represent means ± SEs of three independent replicates. Values which are not significantly different among samples are identified by the same letter (a–c).

**Figure 6 ijms-20-04743-f006:**
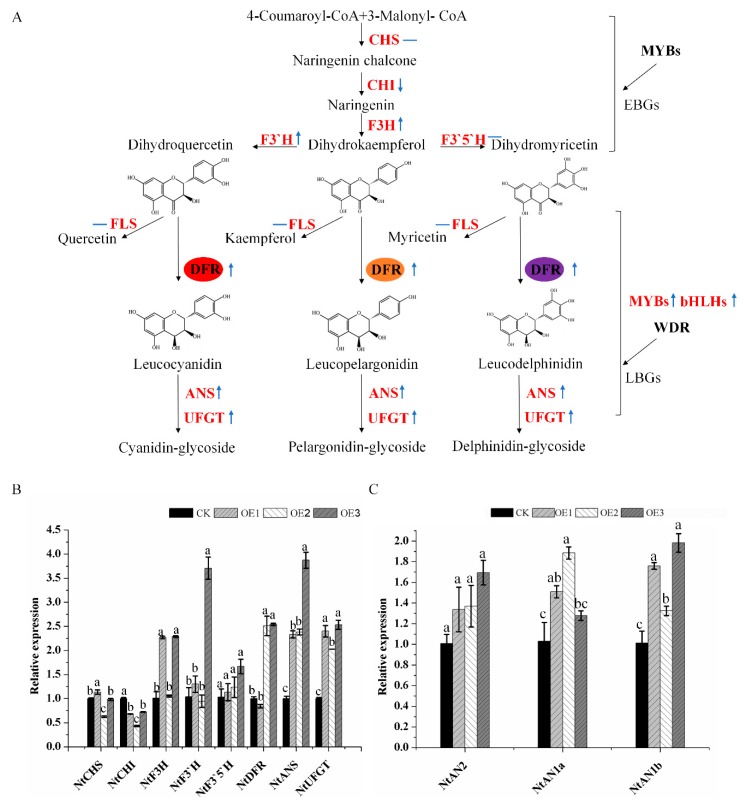
Schematic diagram of the flavonoid biosynthetic pathway leading to anthocyanins and expression profiles of anthocyanin biosynthetic endogenous genes in transgenic tobacco petals. (**A**). A schematic diagram of the flavonoid biosynthetic pathway, in which the dihydroflavonol reductase (DFR) catalyzes the NADPH-dependent reduction of dihydroflavonols to corresponding products. (**B**) Expression profiles of endogenous anthocyanin biosynthetic genes in petals of the CK line and *MaDFR* OE lines (OE1–3). Expression patterns in petals of the early biosynthetic genes (*NtCHS*, *NtCHI*, *F3H*, *NtF3’H* and *NtF3’5’H*) and the late biosynthetic genes (*NtDFR*, *NtANS*, and *NtUFGT*). (**C**) Expression patterns in petals of the regulatory genes *(NtAN2*, *NtAN1a*, and *NtAN1b*). Data represent means ± SEs of three independent experiments. Values which are not significantly different among samples are identified by the same letter (a–c). Abbreviations used are: CHS: chalcone synthase, CHI: chalconeisomerase, F3H: flavanone 3-hydroxylase, F3′H: flavonoid 3′-hydroxylase, F3′5′H: flavonoid 3′5′-hydroxylase, DFR: dihydroflavonol 4-reductase, ANS: anthocyanidin synthase, UFGT: anthocyanidin 3-*O*-glucosyltransferase, MYBs: MYB transcription factors, bHLHs: basic helix-loop-helix proteins, WDR: WD repeat protein, EBGs: early biosynthetic genes, LBGs: late biosynthetic genes. The analyzed genes are highlighted in red. The upward blue arrow represents the upregulation of the gene, the downward blue arrow represents the downregulation of the gene, and the horizontal blue line represents a slight change in gene expression.

**Figure 7 ijms-20-04743-f007:**
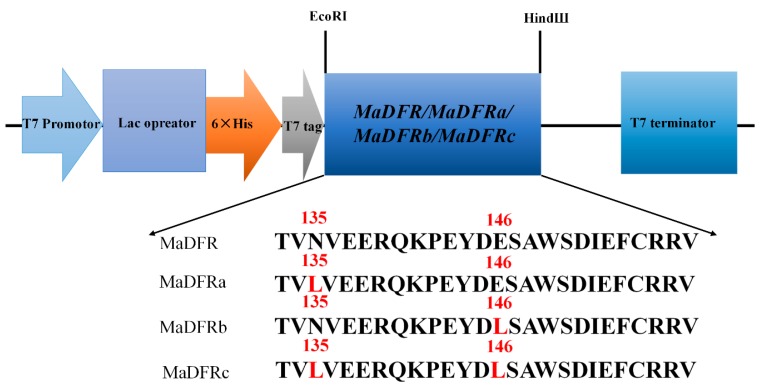
Schematic diagram of the T-DNA region of Site-directed mutagenesis of the region responsible for the substrate specificity of *MaDFR*. Amino acids in red indicate substitutions. MaDFRa, N135 was mutated to L135. MaDFRb, E146 was mutated to L146. MaDFRc, N135 was mutated to L135, and E146 was mutated to L146.

**Table 1 ijms-20-04743-t001:** Enzymatic experiment of MaDFR catalyzing the formation of corresponding products by three substrates.

Substrates	Enzymatic Product	Content (μg/mL) ± SE
DHK	Pg	5.03 ± 0.000262 c
DHQ	Cy	6.45 ± 0.000383 b
DHM	Dp	11.71 ± 0.000493 a

Data represent means ± SEs of three independent replicates, a–c represents a significant difference among different samples.

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
