# Peer review of "Cloning and Functional Characterization of Dihydroflavonol 4-Reductase Gene Involved in Anthocyanidin Biosynthesis of Grape Hyacinth"

_ijms, 2019, doi:10.3390/ijms20194743_

Round 1
Reviewer 1 Report
Dear Editor
This manuscript reports about molecular characterization of DFR in grape hyacinth. Grape hyacinth is a garden bulbous plant that has blue flowers derived from delphinidin (Dp), one of major anthocyanins. DFR is one of the flavonoid (anthocyanin) biosynthesis pathway enzymes, and it is previously reported that the substrate specificity of DFR determines final structure of anthocyanin in some species. In this manuscript, authors performed pigment analysis, gene expression analysis, in vitro enzyme assay and production of MaDFR over expression tobacco plants. Authors concluded MaDFR plays an important role for delphinidin biosynthesis and has some substrate specificity, but the data provided in the manuscript is not enough to lead this conclusion. The topic is interesting, so I recommend adding some additional experiments for publication in International Journal of Molecular Sciences.
<Correlation between MaDFR expression and Dp accumulation>
・Authors mentioned that MaDFR “is highly correlated with anthocyanin accumulation in flower” (L232). Authors showed correlation coefficient only between MaDFR and anthocyanin amount in Fig. 2B (Authors used “r2”, but please use “r”, a correlation coefficient, to show correlation), but MaDFR is just one of candidate determinant factors of Dp biosynthesis. This does not exclude the possibility that other enzyme such as F3’5’H is more important for Dp biosynthesis. Please provide qRT-PCR data for other genes (at least CHS, CHI, F3H, F3’H, F3’5’H, ANS and UFGT, it would be nicer if authors could provide the data for transcription factors) and analyze correlation between anthocyanin accumulation and these gene’s expression level. This manuscript is written in a way that MaDFR would have the most contribution for Dp biosynthesis than other genes, but if other gene is more correlated than DFR, rewrite whole manuscript.
・Another concern is calculation of correlation between Dp and MaDFR expression. How this was calculated? If authors included the scores of roots, bulbs and leaves in this calculation, they are not comparative, so please exclude them.
<DFR substrate specificity>
・In this manuscript, substrate preference of MaDFR is discussed (L31-32: “In summary, MaDFR has a high preference for dihydromyricetin, which could be a powerful candidate gene for genetic engineering for blue flower colour modification.”, L385-389: “In this study,MaDFR preferentially catalyzes DHM and can use all the three dihydroflavonols as substrate, which provides a new sight to understand anthocyanin biosynthesis in grape hyacinth, and it may be an ideal candidate gene to specifically engineering biosynthesis of delphindin-type anthocyanins.”, and L441-443: “Moreover, MaDFR enzyme could utilize all the three types of dihydroflavonols as substrate, and it most prefered dihydromyricetin as substrate to produce leucodelphinidin.”). This discussion is based on the data L258-261: “Quantification analysis of the reaction products indicates that MaDFR showed the strongest preference for DHM as a substrate, followed by DHQ, and lastly DHK; the concentrations of each of these substrates were 0.19, 0.025, and 0.013 mg/ml, respectively.”, but this is insufficient. For quantification, the experiment must be required at least three replications. Please provide the data as the average score ± SE in main figure or table. This data is quite important because only this data indicates DFR substrate specificity.
・L261-262: “Among the mutant MaDFR proteins, MaDFRa and MaDFRb catalyzed all three substrates, whereas MaDFRc showed no activity (data not shown).” I suppose these data are quite important as well in this manuscript. Please show these data in the main figure or table.
<Statistical analysis>
・Authors should use SEs instead of SDs to describe the difference among treatments.
・I think LSD test is not correct to compare more than three treatments. Please use Tukey‒Kramer instead of LSD.
<Other concerns>
・L25-28: This sentence is confusable. Please rewrite to describe the results accurately.
・Materials and methods: Methods for western blotting was not described in the current manuscript. Please add the description for western blot including the antibody sequence used for detection.
・L79-83: Please add temperature and light intensity of growth chamber.
・Fig. 2: Authors should not use a line graph for Fig. 2, because this graph includes not only flower developmental stages but also other tissues, those are not continuous data.
・Fig. 4 and Fig. 5: Fig. 4 and Fig. 5 should be combined into one figure because the data for standard of Fig. 4 is shown in Fig. 5.
・Fig. 6A: Some photos have low lightness and are not good enough to compare. Please provide comparable photos.
・Fig. 7A: Authors are able to draw genes are upregulated only when all three OE lines are significantly higher than CK. So, only ANS and UFGT are upregulated by the current data.
・Authors list (L5) and author contributions (L448-449) are not consistent.
・L592-599: These sentences are for figure legend of Fig. 7A.
・L594: ANS is “Anthocyanidin synthase” not “Anthocyanin synthase”.
・Overall, typing errors were found. Please correct.
Author Response
Dear editor,
We would like to thank you and the reviewer for the critical comments on our manuscript. These valuable comments not only helped us with the improvement of our manuscript, but provided some new ideas for future studies.
Based on the comments we received, careful modifications have been made to the Review 1 manuscript. All changes were marked by using the track changes mode. We hope the revised manuscript will meet your magazine’s standard. Below you will find our point-by-point responses to the reviewers’ comments/ questions.
Best regards,
Sincerely yours,
Yali liu
Northwest A&F University
Yangling Shaanxi 712100
the People's Republic of China
Fax: 0086-029-87082803
Tel: 0086-029-87082803
E-mail: lyl6151@126.com
Response to Reviewer 1 Comments
Point 1: Authors mentioned that MaDFR “is highly correlated with anthocyanin accumulation in flower” (L232). Authors showed correlation coefficient only between MaDFR and anthocyanin amount in Fig. 2B (Authors used “r2”, but please use “r”, a correlation coefficient, to show correlation), but MaDFR is just one of candidate determinant factors of Dp biosynthesis. This does not exclude the possibility that other enzyme such as F3’5’H is more important for Dp biosynthesis. Please provide qRT-PCR data for other genes (at least CHS, CHI, F3H, F3’H, F3’5’H, ANS and UFGT, it would be nicer if authors could provide the data for transcription factors) and analyze correlation between anthocyanin accumulation and these gene’s expression level. This manuscript is written in a way that MaDFR would have the most contribution for Dp biosynthesis than other genes, but if other gene is more correlated than DFR, rewrite whole manuscript.
Response 1: We agree with your point of view. We have corrected it as you suggest. Please see lines 251-252.
We supplemented the experiment according to your opinion, qRT-PCR analysis of the transcriptional expression of CHS, CHI, F3H, F3'H, F3'5'H, DFR, ANS and UFGT. Using the Correlationplot Rectanglel module of Qi MetA software, the r value of these genes and anthocyanin accumulation was recalculated using the Pearson method. The results showed that the correlation between DFR and Dp was significantly higher than that of other genes, as shown in the table below.
|
Anthocyanin
Gene name |
Dp |
Pg |
Cy |
||||||||
|
CHS CHI F3H F3`H F3`5`H DFR ANS UFGT |
|
0.963 -0.366 0.348 0.942 0.975 0.234 0.106 0.273 |
0.652 -0.518 0.690 0.905 0.645 0.632 0.386 0.547 |
qRT-PCR data
|
Different tissues
|
2^-ΔΔCt ± SE
CHI CHS F3H F3`H F3`5`H DFR ANS UFGT |
|||||||
|
R B L S1 S2 S3 S4 S5
|
1.16±0.42 65.49±14.07 3.73±0.89 2.44±0.87 4.91±0.92 519.14±9.57 5186.17±114.95 17844.07±228.6
|
6.89±1.21 1.00±0.05 1.34±0.07 9.76±0.42 35.61±1.67 10.95±0.49 7.82±0.25 11.88±0.21 |
2.14 ±0.13 6.03 ±0.71 1.01±0.10 746.96±30.95 409.69±28.40 1757.96±29.54 1270.31±32.27 1053.61±52.60 |
1.00±0.02 2.94 ±0.47 47.57±2.02 3.11 ±0.38 25.11±1.43 121.73±11.65 107.04±8.79 189.59±5.23 |
93.28±8.46 52.51±6.76 20.28±1.29 1.04±0.20 1.85±0.12 71.41±7.44 123.61±28.02 569.78±28.86 |
1.68±0.47 3.26±0.62 1.02±0.17 41.83±1.11 123.42±4.38 334.19±11.51 212.34±10.47 149.94±5.54 |
1.68±0.11 7.26±0.42 1.03±0.18 91.80±9.31 638.63±44.89 1334.90±43.21 536.44±7.33 516.28±40.33 |
1.01±0.09 0.72±0.09 0.12±0.03 12.46±0.53 7.20±0.31 59.74±2.65 25.45±0.76 25.84±0.65 |
Point 2: Another concern is calculation of correlation between Dp and MaDFR expression. How this was calculated? If authors included the scores of roots, bulbs and leaves in this calculation, they are not comparative, so please exclude them.
Response 2: We are sorry for that we have not described clearly. We add the methods of calculation of correlation between Dp and MaDFR expression. Please see lines 104-107. The Correlationplot Rectanglel module of Qi MetA software was used to calculate the correlation coefficient between MaDFR expression and anthocyanin accumulation of the five stages of flower development using the Pearson method, not including the roots, bulbs and leaves.
Point 3: In this manuscript, substrate preference of MaDFR is discussed (L31-32: “In summary, MaDFR has a high preference for dihydromyricetin, which could be a powerful candidate gene for genetic engineering for blue flower colour modification.”, L385-389: “In this study,MaDFR preferentially catalyzes DHM and can use all the three dihydroflavonols as substrate, which provides a new sight to understand anthocyanin biosynthesis in grape hyacinth, and it may be an ideal candidate gene to specifically engineering biosynthesis of delphindin-type anthocyanins.”, and L441-443: “Moreover, MaDFR enzyme could utilize all the three types of dihydroflavonols as substrate, and it most prefered dihydromyricetin as substrate to produce leucodelphinidin.”). This discussion is based on the data L258-261: “Quantification analysis of the reaction products indicates that MaDFR showed the strongest preference for DHM as a substrate, followed by DHQ, and lastly DHK; the concentrations of each of these substrates were 0.19, 0.025, and 0.013 mg/ml, respectively.”, but this is insufficient. For quantification, the experiment must be required at least three replications. Please provide the data as the average score ± SE in main figure or table. This data is quite important because only this data indicates DFR substrate specificity.Line 89: How was MaDFR retrieved from GenBank? Was the sequence already known before?
Response 3: We agree with your suggestion and provide the data as you suggest. Please see lines 308-310 (Table 1).
Table1 Enzymatic experiment of DFR catalyzing the formation of corresponding products by three substrates
|
Substrates Enzymatic product Content(mg/ml) ± SE |
|
DHK Pg 0.0050314 ± 0.00026205c
DHQ Cy 0.0064541 ± 0.000383953b
DHM Dp 0.0117194 ± 0.000493094a |
We are sorry for our unclear statement. This sentence is said that the DFR proteins of other species come from the GenBank, not MaDFR. MaDFR is a new gene we cloned from the grape hyacinth.
Point 4: L261-262: “Among the mutant MaDFR proteins, MaDFRa and MaDFRb catalyzed all three substrates, whereas MaDFRc showed no activity (data not shown).” I suppose these data are quite important as well in this manuscript. Please show these data in the main figure or table.
Response 4: We agree with your suggestion and add the figure as you suggest in the manuscript. Place see lines 310-311, Figure 5.
Point 5: Authors should use SEs instead of SDs to describe the difference among treatments.
Response 5: We have corrected it as you suggest. Place see lines 118, 353, 373.
Point 6: I think LSD test is not correct to compare more than three treatments. Please use Tukey‒Kramer instead of LSD.
Response 6: We have corrected it as you suggest. Please see lines 196; Figure2B, C, D, E, F in line 109; Figure6B, C in line 346; Figure7B, C in line 362.
Point 7: L25-28: This sentence is confusable. Please rewrite to describe the results accurately.
Response 7: We have rewrite it as you suggested. To further verify the substrate-specific selection domains of MaDFR, the assay of amino acid substitutions was conducted. The activity of MaDFR did not affect whenever N135 or E146 site was mutated. Please see lines 25-27.
Point 8: Materials and methods: Methods for western blotting was not described in the current manuscript. Please add the description for western blot including the antibody sequence used for detection.
Response 8: We add the methods for western blotting as you suggested. Please see lines 143-147.
Point 9: L79-83: Please add temperature and light intensity of growth chamber.
Response 9: We add the information in lines 80-81.
Point 10: Authors should not use a line graph for Fig. 2, because this graph includes not only flower developmental stages but also other tissues, those are not continuous data.
Response 10: We have corrected it as you suggest. Please see Figure2 in lines 109-110.
Point 11: Fig. 4 and Fig. 5: Fig. 4 and Fig. 5 should be combined into one figure because the data for standard of Fig. 4 is shown in Fig. 5.
Response 11: We agree with your point of view. We have corrected it as you suggest, Please see Figure 4 in lines 286-287.
Point 12: Fig. 6A: Some photos have low lightness and are not good enough to compare. Please provide comparable photos.
Response 12: We have corrected it as you suggest. Please see Figure6 in lines 346-347.
Point 13: Fig. 7A: Authors are able to draw genes are upregulated only when all three OE lines are significantly higher than CK. So, only ANS and UFGT are upregulated by the current data.
Response 13: We have corrected it as you suggest. Please see lines 357, 423-424.
Point 14: Authors list (L5) and author contributions (L448-449) are not consistent.
Response 14: We have corrected it as you suggest. Please see lines 466-467.
Point 15: L592-599: These sentences are for figure legend of Fig. 7A.
Response 15: We are sorry for a simple mistake. We have corrected it as you suggest. Please see lines 373-380.
Point 16: L594: ANS is “Anthocyanidin synthase” not “Anthocyanin synthase”.
Response 16: We are sorry for a mistake in our writing. We have corrected “Anthocyanin synthase” as “Anthocyanidin synthase”. Please see line 375.
Point 17: Overall, typing errors were found. Please correct.
Response 17: We have corrected it as you suggest.
Reviewer 2 Report
The authors used RNA-Seq data from Muscari armeniacum to identify and clone the cDNA of a Muscari aucheri dihydroflavonol 4-reductase (MaDFR) gene.
Site directed mutagenesis of the MaDFR gene changed amino acid positions N135 and E146 of the encoded protein to L135/E146, N135/L146 and L135/L146, respectively.
Wild type and the three mutant MaDFR genes were overexpressed in E. coli. In vitro assays with all three dihydroflavonols (dihydrokaempferol, dihydroquercetin, and dihydromyricetin) as substrates were performed to confirm the expected enzymatic activity of the encoded MaDFR proteins and to characterize their substrate specificity.
It could be shown, that the at least N135 or E146 of MaDFR are necessary for its enzymatic function.
The wild type MaDFR gene was ectopically overexpressed in N. tabacum, leading to altered transcript levels of enzymes involved in the flavonoid biosynthetic pathway and finally to an increase in flower anthocyanin content.
For the reader it is difficult to work through this paper. Though the authors present many experimental data, many details necessary for understanding the experimental design are missing or wrong. The logical order and language have to be improved.
Before the paper can be published, the following main points have to be corrected:
The supplemental figures and tables are completely missing. This information has to be reviewed before publication. The MaDFR nucleotide sequence has to be revealed. The indicated accession MK937098 does not exist. It has to be clearly mentioned in the main text (and not only somewhere in legends to figures) that the native and mutated MaDFR proteins overexpressed in E. coli carry His-tags, and at which position the tags are. A map of the pCAMBIA2300 derivative (at least T-DNA) should be provided. The identity of the N. tabacum empty vector control is unclear. In line 334 the authors write: “control (CK) line transformed with pET28a”. pET28a is an E. coli vector and does neither carry plant transcription regulation signals nor T-DNA left/right border signals. Agrobacterium mediated transformation with this plasmid is not possible. The N. tabacum experiments should be done with an appropriate negative control. Figure 3 indicates RB and LB for the constructs. The legend to the figure lacks almost all necessary information. From the main text the reader would conclude that the figure shows derivatives of plasmid pET28a. This plasmid does not contain any RB/LB. In connection with point 5 it seems that the authors do not know how to use the plasmids they are working with. The authors should show the experimental data that support the results for the mutant MaDFR proteins (lines 261 and 262). The authors should explain abbreviations when they are used the first time (not in line 53, but already in line 42) or in one paragraph (and not split between lines 453-456 and 592-599). NtAn (line 343) should be explained somewhere.
Minor points are:
Lines 85-87: Please explain the cloning of the gene in more detail. Why was this gene chosen? What enzyme/chemicals were used to clone the cDNA?
Line 89: How was MaDFR retrieved from GenBank? Was the sequence already known before?
Lines 104-106: The data points in Fig. 2B should not be connected by a line. Either a bar graph is presented or the authors separate R, B, L (different organs at one time point) from S1-5 (same organ at different time points).
Line 119: E. coli was transformed with the plasmid! This error appears later in the manuscript again.
Lines 121-122: what is kana?
Line 127: Sonication using what instrument, what power output?
Line 155: Is the vector name p2300-35SMaDFR or p2300-MaDFR?
Line 156-157: Please mention the transformation method here.
Line 195: What are maDFR from Muscari?
Lines 262-264: No, these results suggest that at least one of the residues is necessary.
Line 274: Lane 4, do the authors mean the pellet fraction?
Line 322: “data not shown”: Did the authors determine the copy number of T-DNA insertions in their chosen lines?
Lines 379, 395, 435: Please correct sentences.
Lines 537-539: First names and family names are confused.
Author Response
Dear editor,
We would like to thank you and the reviewer for the critical comments on our manuscript. These valuable comments not only helped us with the improvement of our manuscript, but provided some new ideas for future studies.
Based on the comments we received, careful modifications have been made to the Review 1 manuscript. All changes were marked by using the track changes mode. We hope the revised manuscript will meet your magazine’s standard. Below you will find our point-by-point responses to the reviewers’ comments/ questions.
Best regards,
Sincerely yours,
Yali liu
Northwest A&F University
Yangling Shaanxi 712100
the People's Republic of China
Fax: 0086-029-87082803
Tel: 0086-029-87082803
E-mail: lyl6151@126.com
Response to Reviewer 2 Comments
Point 1: The supplemental figures and tables are completely missing. This information has to be reviewed before publication. The MaDFR nucleotide sequence has to be revealed. The indicated accession MK937098 does not exist. It has to be clearly mentioned in the main text (and not only somewhere in legends to figures) that the native and mutated MaDFR proteins overexpressed in E. coli carry His-tags, and at which position the tags are. A map of the pCAMBIA2300 derivative (at least T-DNA) should be provided. The identity of the N. tabacum empty vector control is unclear. In line 334 the authors write: “control (CK) line transformed with pET28a”. pET28a is an E. coli vector and does neither carry plant transcription regulation signals nor T-DNA left/right border signals. Agrobacterium mediated transformation with this plasmid is not possible. The N. tabacum experiments should be done with an appropriate negative control. Figure 3 indicates RB and LB for the constructs. The legend to the figure lacks almost all necessary information. From the main text the reader would conclude that the figure shows derivatives of plasmid pET28a. This plasmid does not contain any RB/LB. In connection with point 5 it seems that the authors do not know how to use the plasmids they are working with. The authors should show the experimental data that support the results for the mutant MaDFR proteins (lines 261 and 262). The authors should explain abbreviations when they are used the first time (not in line 53, but already in line 42) or in one paragraph (and not split between lines 453-456 and 592-599). NtAn (line 343) should be explained somewhere.
Response 1:
We are very sorry that we made such a serious mistake. We have submitted these important data again.
We revealed the MaDFR nucleotide sequence nucleotide. Please see Supplementary figure S1.
We have submitted the MaDFR nucleotide sequence data to GenBank, the scheduled release date is Sep 10, 2019.
We have added the information of the native and mutated MaDFR proteins overexpressed in E. coli carry His-tags, and at which position the tags are. Please see lines 124, 132-133.
We provide the map of the T-DNA of the pCAMBIA2300. Please see Supplementary figure S3
We are very sorry for a mistake in our writing. We have corrected it, control (CK) line transformed with pCAMBIA2300 empty vector, please see lines 176, 349.
We are sorry for a simple mistake. We have corrected it as you suggest. Please see line 126, Figure 3. We have added the necessary information in legend of Figure 3, please lines 127-130.
We agree with your suggestion and provide the data as you suggest. Please see lines 310-311, Figure 5.
We are sorry for a simple mistake. We have corrected it as you suggest. Please see lines 42, 52-53. L592-599: These sentences are for figure legend of Fig. 7A. We have corrected it, please see lines 373-380.
We are sorry for our unclear statement. We have corrected it as you suggest. Please see line 356-357.
Point 2: Lines 85-87: Please explain the cloning of the gene in more detail. Why was this gene chosen? What enzyme/chemicals were used to clone the cDNA?.
Response 2: We are sorry for our unclear statement. We add the cloning of the gene in more detail in Lines 84-89.
The previous transcriptome sequencing and metabolome of our research group proposed two points: 1, DFR sequences showed significantly higher levels of gene transcripts in blue flowers than in white flowers. 2, No products of the Del synthesis route that occur after dihydromyricetin (the substrate for the DFR enzyme) were detected in white flowers. These results suggest that DFR was the most likely target gene for the lack of Del in white flowers. So, we chose this gene to continue research.
The PrimeSTAR® HS DNA Polymerase (Japanese, Takara) was used to clone the cDNA.
Point 3: Line 89: How was MaDFR retrieved from GenBank? Was the sequence already known before?
Response 3: We are sorry for our unclear statement. This sentence is said that the DFR proteins of other species come from the GenBank, MaDFR is our newly cloned. There was no the sequence before.
Point 4: Lines 104-106: The data points in Fig. 2B should not be connected by a line. Either a bar graph is presented or the authors separate R, B, L (different organs at one time point) from S1-5 (same organ at different time points).
Response 4: We have corrected it as you suggest. Please see lines 109-110, Figure 2B, C, D, E, F.
Point 5: Line 119: E. coli was transformed with the plasmid! This error appears later in the manuscript again.
Response 5: We are sorry for our unclear statement. We have corrected it as you suggest. The Escherichia coli strain BL21 (DE3) was transformed with the empty pET28a vector and recombinant MaDFR plasmids. Please see lines 132-133. Agrobacterium tumefaciens strain GV3101 was transformed with the p2300-35SMaDFR and the empty pCAMBIA2300 by electroporation. Please see lines 175-177.
Point 6: Lines 121-122: what is kana?
Response 6: Kana is the abbreviation of Kanamycin, we have corrected it. Please see lines 134-135.
Point 7: Line 127: Sonication using what instrument, what power output?
Response 7: We add the instrument, and power output in Line 141. Noise Isolating Chamber (SCIENTZ, China), 200w.
Point 8: Line 155: Is the vector name p2300-35SMaDFR or p2300-MaDFR?
Response 8: We are sorry for our unclear statement. Yes, the recombinant plasmid is p2300-35SMaDFR, we corrected p2300-MaDFR and p35SMaDFR as p2300-35SMaDFR. Please see lines 175-176.
Point 9: Line 156-157: Please mention the transformation method here.
Response 9: We add the transformation method in line 178.
Point 10: Line 195: What are maDFR from Muscari?
Response 10: We are sorry for not clearly stating the sentence. We have corrected it as alignment of protein sequence and anlysis of phylogenetic tree of MaDFR. Please see line 216.
Point 11: Lines 262-264: No, these results suggest that at least one of the residues is necessary
Response 11: We agree with your point of view. We have corrected it as you suggest, These results suggest that at least one of the residues (asparagine [N135] and glutamic acid residues [E146]) is necessary for the catalytic activity of the MaDFR enzyme. Please see lines 282-283.
Point 12: Line 274: Lane 4, do the authors mean the pellet fraction?
Response 12: Yes.
Point 13: Line 322: “data not shown”: Did the authors determine the copy number of T-DNA insertions in their chosen lines?
Response 13: We are very sorry that this experiment was not done. But, 11 independent transgenic lines with different with different degrees of of color deepening, which indirectly indicates that the transgenic results are credible, and qPCR data also can support this conclusion.
Point 14: Lines 379, 395, 435: Please correct sentences.
Response 14: We are sorry for not clearly stating the sentence. We have corrected it as you suggest. Please see lines 399-400. In most plant species, the 134th amino acid was N or D, and the 145th was glutamic acid (E).
Please see lines 413-414.These results indicated that there might be other amino acid binding site influencing the specificity substrate of MaDFR.
Please see lines 452-455. The presence of Pg derivatives in blue petals and the catalytic activity of MaDFR for DHK indicated that there must be the anthocyanin biosynthetic pathway based on red Pg in this genus, hinting other functional genes responsible for the loss of red Pg-based pigments in grape hyacinth.
Point 15: Lines 537-539: First names and family names are confused.
Response 15: We are sorry for a simple mistake. We have corrected it as you suggest. Please see lines 561-564.
Round 2
Reviewer 1 Report
In this revision, authors corrected some sentences and figures, and added some data. It seems that findings of this manuscript is getting clearer, still, there are some points to be revised for publication in International Journal of Molecular Sciences.
>We supplemented the experiment according to your opinion, qRT-PCR analysis of the transcriptional expression of CHS, CHI, F3H, F3'H, F3'5'H, DFR, ANS and UFGT. Using the Correlationplot Rectanglel module of Qi MetA software, the r value of these genes and anthocyanin accumulation was recalculated using the Pearson method. The results showed that the correlation between DFR and Dp was significantly higher than that of other genes, as shown in the table below.
|
Anthocyanin Gene name |
Dp |
Pg |
Cy |
|
CHS CHI F3H F3`H F3`5`H DFR ANS UFGT |
0.399 -0.421 0.812 0.816 0.415 0.850 0.649 0.73 |
0.963 -0.366 0.348 0.942 0.975 0.234 0.106 0.273 |
0.652 -0.518 0.690 0.905 0.645 0.632 0.386 0.547 |
・Please add this table (correlation of Pg, Cy and Dp and anthocyanin biosynthetic gene expression levels) in the supplemental data, and mention it in the main text.
・L88: “Japanese, Takara” is not following the format.
・L134, 135: 50mg/mL kanamycin is final concentration? It seems very high.
・Table 1: In the title of the table, “DFR” should be “MaDFR”.
・Table 1: Please mind significant digits. It would be better to use µg/mL±SE in this table.
・Fig. 6: If authors have not determined the copy number of T-DNA insertions in the chosen lines, it would be better to mention it in the main text.
・Fig. 6B: Although authors used SE instead of SD in this time, but still the length of bars looks the same to the first manuscript. Please check all figures that all bars are correct.
・L472: Abbreviations does not contain whole abbreviations in this manuscript.
・L487-488: Reference [3] is not following the format.
・There are some errors in L42, L46, L176, L281 and L356.
Author Response
Dear editor,
We would like to thank you and the reviewer for the critical comments on our manuscript again. Thank you for giving us the opportunity to resubmit this manuscript for possible publication in International Journal of Molecular Sciences. Here below is our description on revision according to the reviews' and editor's comments
We deeply appreciate your consideration.
Response to Reviewer 1 Comments
Point1 We supplemented the experiment according to your opinion, qRT-PCR analysis of the transcriptional expression of CHS, CHI, F3H, F3'H, F3'5'H, DFR, ANS and UFGT. Using the Correlationplot Rectanglel module of Qi MetA software, the r value of these genes and anthocyanin accumulation was recalculated using the Pearson method. The results showed that the correlation between DFR and Dp was significantly higher than that of other genes, as shown in the table below.
・Please add this table (correlation of Pg, Cy and Dp and anthocyanin biosynthetic gene expression levels) in the supplemental data, and mention it in the main text.
Response 1: We agree with your point of view. We have added this table in the supplemental data as you suggest. Please see Supplementary Table S2.
Point2・L88: “Japanese, Takara” is not following the format.
Response 2: We have corrected it as you suggest. Please see in lines 88, 90, 99, 100, 102, 104, 144, 147, 148, 191 and 192.
Point3・L134, 135: 50mg/mL kanamycin is final concentration? It seems very high.
Response 3: We are sorry for a mistake in our writing. We have corrected 50mg/mL as 50 ug/mL. Please see in lines 137,138.
Point4・Table 1: In the title of the table, “DFR” should be “MaDFR”.
Response 4: We have corrected it as you suggest. Please see in line 313.
Point5・Table 1: Please mind significant digits. It would be better to use µg/mL±SE in this table.
Response 5: We have corrected it as you suggest. Please see Table1 in lines 312-316.
Point6・Fig. 6: If authors have not determined the copy number of T-DNA insertions in the chosen lines, it would be better to mention it in the main text.
Response 6: We have have added this information as you suggest. Please see in line 341-342.
Point7・Fig. 6B: Although authors used SE instead of SD in this time, but still the length of bars looks the same to the first manuscript. Please check all figures that all bars are correct.
Response 7: We are sorry for a mistake and have corrected it as you suggest. Please see Fig.6B. We have cheked all figures that all bars and the others were corrected.
Point8・L472: Abbreviations does not contain whole abbreviations in this manuscript.
Response 8: We have corrected it as you suggest. Please see in lines 485-492.
Point9・L487-488: Reference [3] is not following the format.
Response 9: We have corrected it as you suggest. Please see in line 504-505.
Point10・There are some errors in L42, L46, L176, L281 and L356.
Response 10: We have corrected it as you suggest. Please see in lines 43, 47, 178-180, 285, 364 and 433-435.
Reviewer 2 Report
Most of the points criticized in the previous review have been corrected.
Some points have to be clarified before publication:
1) In lines 259/260 the authors write: "The MaDFRs protein was identified at the predicted molecular mass of 41 kDa by western blotting (Figure 4B and Supplementary figure S2D)."
The figures mentioned show a protein with a molecular mass even bigger than 45 kDa. The MaDFR sequence submitted by the authors as GenBank MK937098 indicates a protein with a molecular mass of 40.9 kDa, the addition of 6 His residues would lead to a molecular mass of 41.8 kDa. Before publication the authors have to explain the difference between the expected molecular mass and the molecular mass experimentally obtained for MaDFR.
2) Line 91: Only the amino acid sequences of other DFR proteins were retrieved from GenBank, not the one of MaDFR.
3) With the corrections made new spelling errors were introduced into the paper like in lines 141 and 178. A spell check is required.
Author Response
Dear editor,
We would like to thank you and the reviewer for the critical comments on our manuscript again. Thank you for giving us the opportunity to resubmit this manuscript for possible publication in International Journal of Molecular Sciences. Here below is our description on revision according to the reviews' and editor's comments
We deeply appreciate your consideration.
Response to Reviewer 2 Comments
Comments and Suggestions for Authors Most of the points criticized in the previous review have been corrected.
Some points have to be clarified before publication:
1) In lines 259/260 the authors write: "The MaDFRs protein was identified at the predicted molecular mass of 41 kDa by western blotting (Figure 4B and Supplementary figure S2D)."
The figures mentioned show a protein with a molecular mass even bigger than 45 kDa. The MaDFR sequence submitted by the authors as GenBank MK937098 indicates a protein with a molecular mass of 40.9 kDa, the addition of 6 His residues would lead to a molecular mass of 41.8 kDa. Before publication the authors have to explain the difference between the expected molecular mass and the molecular mass experimentally obtained for MaDFR.
Response 1: We are very grateful for your question and give a reasonable explanation.
On one hand, the molecular mass experimentally obtained should be the predicted molecular weight of the MaDFR protein plus the molecular weight of the 6HIS and T7 Tag proteins, ie 40.9 + 0.8 + 2.4 = 44.1 kDa. On the other hand, the size of the protein Marker is not absolutely standard in different concentrations of SDS-PAGE gel, and It is normal for the predicted protein size to fluctuate in the upper and lower ranges of the Marker band, and does not affect the results of the experiment.
2) Line 91: Only the amino acid sequences of other DFR proteins were retrieved from GenBank, not the one of MaDFR.
Response 2: We have corrected it as you suggest. Please see in line 94-95.
3) With the corrections made new spelling errors were introduced into the paper like in lines 141 and 178. A spell check is required
Response 3: We have corrected it as you suggest. Please see in lines 143-144, 180.